# Taxonomizing local versus global structure in neural network loss landscapes

**Yaoqing Yang**[1], **Liam Hodgkinson**[1,2], **Ryan Theisen**[1], **Joe Zou**[1],
**Joseph E. Gonzalez**[1], **Kannan Ramchandran**[1], **Michael W. Mahoney**[1,2]
[1] University of California, Berkeley
[2] International Computer Science Institute
{yqyang, liam.hodgkinson, theisen, joezou, jegonzal, kannanr,
mahoneymw}@berkeley.edu

## Abstract

Viewing neural network models in terms of their loss landscapes has a long history
in the statistical mechanics approach to learning, and in recent years it has received
attention within machine learning proper. Among other things, local metrics
(such as the smoothness of the loss landscape) have been shown to correlate with
global properties of the model (such as good generalization performance). Here, we
perform a detailed empirical analysis of the loss landscape structure of thousands of
neural network models, systematically varying learning tasks, model architectures,
and/or quantity/quality of data. By considering a range of metrics that attempt
to capture different aspects of the loss landscape, we demonstrate that the best
test accuracy is obtained when: the loss landscape is globally well-connected;
ensembles of trained models are more similar to each other; and models converge
to locally smooth regions. We also show that globally poorly-connected landscapes
can arise when models are small or when they are trained to lower quality data;
and that, if the loss landscape is globally poorly-connected, then training to zero
loss can actually lead to worse test accuracy. Our detailed empirical results shed
light on phases of learning (and consequent double descent behavior), fundamental
versus incidental determinants of good generalization, the role of load-like and
temperature-like parameters in the learning process, different influences on the loss
landscape from model and data, and the relationships between local and global
metrics, all topics of recent interest.

## 1 Introduction

Among the many approaches to understanding the behavior of neural network (NN) models, the study
of their loss landscapes [1, 2] has proven to be particularly fruitful. Indeed, analyzing loss landscapes
has helped shed light on the workings of many popular techniques, including large-batch training
[3, 4], adversarial training [5], residual connections [6], and BatchNorm [7]. One particular concept
of recent interest is the so-called *sharpness* of local minima [3, 5, 8–10]. While sharpness can be
measured by first-order sensitivity measures, such as the Jacobian or Lipschitz constant, it is more
appropriately measured by second-order sensitivity measures, typically via the Hessian spectrum
[11]. It has been observed that in some cases NNs generalize well when they converge to a relatively
flat, i.e., non-sharp, local minimum [3].

While such local sharpness measures can provide insight, their focus on the local geometry of the loss
landscape neglects the *global* structure of the loss landscape (namely, precisely the sort of structure
that statistical mechanics approaches to learning aim to quantify [12, 13]). Indeed, it is well-known
that existing sharpness-based metrics can be altered (trivially) by reparameterization tricks or (more
interestingly) by taking algorithmic steps which have the effect of changing the local structures on the
loss landscape [5, 8, 14]. For example, [5] shows that adversarial training can decrease the magnitude

35th Conference on Neural Information Processing Systems (NeurIPS 2021).

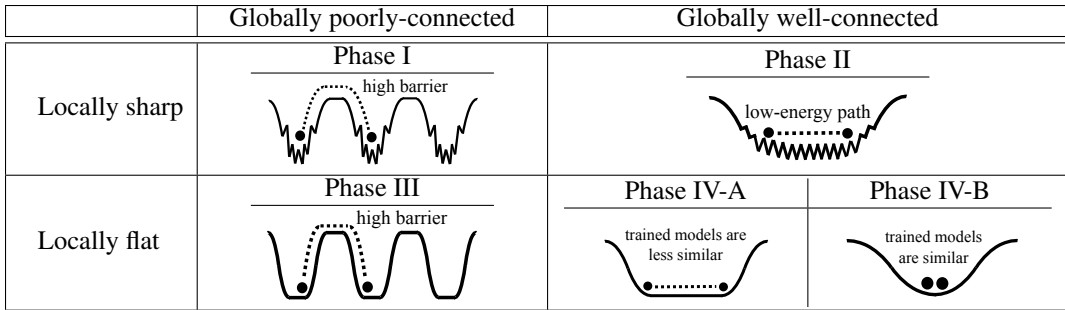

| | Globally poorly-connected | Globally well-connected | |
|---|---|---|---|
| Locally sharp | Phase I
high barrier | Phase II
low-energy path | |
| Locally flat | Phase III
high barrier | Phase IV-A
trained models are
less similar | Phase IV-B
trained models
are similar |

Figure 1: **(Caricature of different types of loss landscapes).** Globally well-connected versus globally poorly-connected loss landscapes; and locally sharp versus locally flat loss landscapes. Globally well-connected loss landscapes can be interpreted in terms of a global "rugged convexity"; and globally well-connected and locally flat loss landscapes can be further divided into two sub-cases, based on the similarity of trained models.

of Hessian eigenvalues and bias the model towards a locally smooth area, even though adversarial training can reduce clean test accuracy [15]. Similarly, [14] shows that Hessian eigenvalues become smaller with reduced $\ell_2$ regularization, even though increased $\ell_2$ regularization is known to reduce overfitting and improve training, if used properly. More general considerations would suggest (and indeed our own empirical results, e.g., as reported in Figure 7, demonstrate) that by training to data with noisy labels, one can find models that generalize poorly and yet simultaneously lie in very "flat" regions of the loss landscape, with small Hessian eigenvalues, and vice versa. These observations (and other observations we describe below) indicate that the previously-observed empirical correlation between very local metrics like sharpness and more global properties like generalization performance may be correlative and not causative, i.e., they may be due to the confounding factor that results in the published literature are on reasonably-good models trained to reasonably-good data, rather than due to some fundamental properties of deep NNs. They also raise the question of how to capture more global properties of the loss landscape.

Motivated by these considerations, we are interested in understanding local properties/structure versus global properties/structure of the loss landscape of realistic NN models. While similar ideas underlie work that adopts a statistical mechanics perspective [2, 12, 13, 16], here we are interested in adopting an operational machine learning (ML) perspective, where we employ metrics that have been used within ML as "experimental probes" to gain insight into local versus global properties. To do so, we employ the following metrics.

- First, we consider *Hessian-based metrics*, including the largest eigenvalue and the trace of the Hessian. These metrics try to capture *local curvature properties of the loss landscape* [11].
- Second, we use *mode connectivity* [17, 18]—in particular, the connectivity between trained models. This metric tries to capture *how well-connected different local minima are* to each other.
- Third, we use *CKA similarity* [19] to capture *a correlation-like similarity* between the outputs of different trained models. Averaging the CKA over several pairs of models can be thought of as approximating so-called overlap integrals frequently appearing in statistical mechanics [12, 13, 20].

We have considered many other metrics, but these three seem to be particularly useful for identifying global structure versus local structure in loss landscapes. Informally, mode connectivity, as its name suggests, captures *connectivity*, where well-connected models exhibit a single "rugged basin" with low-energy / low-loss, potentially non-linear, paths through the loss landscape, (i.e., continuous chains of models) all achieving a small loss value. We expect this property to be important since the connectivity of local minima indicates efficiency of the training dynamics to explore the loss landscape, without becoming stuck at saddle points or in a "bad" local minimum. Similarly, CKA captures *similarity*, where an ensemble of good models will produce roughly similar outputs. These two types of metrics are different and complementary; and both of them are very different than Hessian-based metrics, which clearly capture much more local information.

Here we briefly summarize our main contributions.

- We design an experimental setup based on two control parameters, a *temperature-like* parameter that correlates with the magnitude of SGD noise during training, e.g., batch size (in most figures),

learning rate, or weight decay, and a *load-like* parameter that measures the relationship between model size and data quantity and/or quality, e.g., the amount of data, size of intermediate layers, amount of exogenously-introduced label noise, etc. By training thousands of models, under a variety of settings, and by measuring local and global metrics of the loss landscape, we identify four distinct phases in temperature-load space, with relatively sharp transitions between them.

- Using global connectivity (measured by mode connectivity) and local flatness (measured by the Hessian), we taxonomize loss landscapes into four categories, which are pictorially represented in Figure 1, labelled Phase I through Phase IV. For reasons observed in our empirical results in Section 3, it is often convenient to further divide Phase IV into two subcategories, depending on whether the trained models produce similar representations (as measured by CKA similarity). If the loss landscape satisfies the first property, we say it is *globally well-connected*; and if the loss landscape *also* satisfies the second property, we say it is *globally nice*. Depending on whether the Hessian eigenvalues are large or small, we say the loss landscape is *locally sharp* or *locally flat*.

- Based on these results, as well as measured model quality, e.g., test accuracy, we empirically demonstrate that the global (but not necessarily local) structure of a loss landscape is well-correlated with good generalization performance, and that the best generalization occurs in the phase associated with a locally flat, globally nice loss landscape. We demonstrate these results on a range of computer vision and natural language processing benchmarks (CIFAR-10, CIFAR-100, SVHN, and IWSLT 2016 De-En) and various models (ResNet, VGG, and Transformers). We also vary the amount of data, the number of noisy labels, etc., to study both the effect of the quantity of data and the quality of data on changing the loss landscape.

- We observe the well-known double descent phenomenon [21, 22] in our experiments, which exhibits itself as a "bad fluctuation" between the different phases (e.g., see the transition that separates Phase I and II from Phase III and IV in Figure 4a). Our empirical observations on double descent corroborates recent theoretical analysis [23, 24], which views the phenomenon as a consequence of a transition between qualitatively different phases of learning [13].

Computing connectivity and similarity requires comparing multiple distinct models. This could be computationally expensive, especially if model training is expensive. For many reasonably-sized models, however, the metrics we consider are sufficiently tractable so as to be useful, e.g., during model training. Moreover, the phase transitions and the metrics that we use to determine the phases lead to practical tools that can diagnose typical failure modes in training, which we will discuss towards the end of the paper. In this short conference version, we focus on the main message, and we provide a much more thorough discussion on prior work in the full version online [25], in which we also talk about related papers in the study of loss landscapes and statistical mechanics of learning. In order that our results can be reproduced and extended, we have open-sourced our code.[1]

## 2 Setup

In the sequel, we consider training a NN $f_\theta : \mathbb{R}^{d_{\text{in}}} \to \mathbb{R}^{d_{\text{out}}}$, with trainable parameters $\theta$, to a dataset consisting of $n$ datapoint/label pairs $S_{\text{train}} = \{(\boldsymbol{x}_1, y_1), \ldots, (\boldsymbol{x}_n, y_n)\}$. Our nominal training objective is to minimize a loss function of the form

$$\mathcal{L}(\theta) = \frac{1}{n} \sum_{(\boldsymbol{x}, y) \in S_{\text{train}}} \ell(f_\theta(\boldsymbol{x}), y) + \lambda \|\theta\|_2^2. \tag{1}$$

Here $\ell$ is a loss function, typically chosen to be the cross entropy loss. The parameter $\lambda$ is the weight decay parameter, which controls the level of $\ell_2$ regularization. We consider optimizing NN models using standard minibatch SGD, with iterates of the form

$$\theta \leftarrow \theta - \eta \tilde{\boldsymbol{g}}(\theta), \quad \tilde{\boldsymbol{g}}(\theta) = \frac{1}{B} \sum_{j=1}^{B} \nabla_\theta \ell(f_\theta(\boldsymbol{x}_{i_j}), y_{i_j}), \tag{2}$$

where $\eta$ is the learning rate, $0 < B \leq n$ is the batch size, and the indices $i_1, \ldots, i_B$ of each minibatch are sampled without replacement from $\{1, \ldots, n\}$. For classification tasks, we consider also the training/testing accuracy, which is simply the fraction of correctly classified points, $\text{acc}_{\text{train}}(\theta) = \frac{1}{n} \sum_{(\boldsymbol{x}, y) \in S_{\text{train}}} \mathbf{1}(f_\theta(\boldsymbol{x}) = y)$, and similarly for $\text{acc}_{\text{test}}(\theta)$ on a given test set $S_{\text{test}}$.

---

[1]https://github.com/nsfzyzz/loss_landscape_taxonomy

We now briefly introduce the main metrics and control parameters which we will consider.

**Temperature and load.** In the sequel, a *load-like parameter* of a loss landscape refers to some quantity related to the amount and/or quality of data, relative to the size of the model. Specifically, we vary either i) model size (e.g., width, which captures the size of an internal representation of the data), for fixed training set size $n$, ii) training set size $n$, for fixed model size, or iii) the "quality" of training data, which is varied by randomizing a fraction $\alpha$ of the training labels. Each of these control parameters *directly* induces a different loss landscape by changing the data $S_{\text{train}}$ and/or architecture $f_\theta$ for which the loss $\mathcal{L}(\theta)$ is being computed. For example, we expect that increasing width will result in a smoother loss landscape [26]; we shall see this effect with CKA similarity in the transition from Phase IV-A to IV-B.

The second control parameter we vary in our experiments is a *temperature-like parameter*, representing the amount of noise introduced in the SGD iterates (2). Most commonly, we take this to be the batch size $B$, although we will also use the learning rate $\eta$ and the weight decay parameter $\lambda$. Increasing temperature corresponds to smaller batch size, and large learning rate or weight decay. Varying the temperature does not directly define a different loss function $\mathcal{L}(\theta)$, but rather it *indirectly* induces a different *effective loss function*. This is because, at different temperatures, the iterates of SGD concentrate on different regions of the loss landscape. Due to the noise in the stochastic optimization, the training dynamics may not be able to "see" certain features of the loss landscape.

**CKA similarity.** To measure the similarity of two NN representations, we use the *centered kernel alignment* (CKA) metric, proposed in [19]. For a NN $f_\theta$, let $F_\theta = [f_\theta(\boldsymbol{x}_1) \quad \cdots \quad f_\theta(\boldsymbol{x}_m)]^\top \in \mathbb{R}^{m \times d_{\text{out}}}$ denote the concatenation of the outputs[2] of the network over a set of $m$ randomly sampled datapoints. Then the (linear) CKA similarity between two parameter configurations $\theta, \theta'$ is given by

$$s(\theta, \theta') = \frac{\text{Cov}(F_\theta, F_{\theta'})}{\sqrt{\text{Cov}(F_\theta, F_\theta)\text{Cov}(F_{\theta'}, F_{\theta'})}}, \tag{3}$$

where for $X, Y \in \mathbb{R}^{m \times d}$, we define $\text{Cov}(X, Y) = (m-1)^{-2}\text{tr}(XX^\top H_m YY^\top H_m)$, and $H_m = I_m - m^{-1}\mathbf{1}\mathbf{1}^\top$ is the centering matrix. The CKA similarity is known to be an effective way to compare the overall representations learned by two different trained NNs [19]. Rather than computing the similarity directly on the original training points, we measure CKA on a perturbed training set comprised of Mixup samples [27]; this can reduce trivial similarity that occurs when the models are trained to exactly or approximately zero training error. See also Appendix A.4.1 in the full paper for the ablation study on different perturbed training sets.

**Mode connectivity.** For two parameter configurations $\theta, \theta'$, computing mode connectivity involves finding a *low-energy curve* $\gamma(t)$, $t \in [0, 1]$, for which $\gamma(0) = \theta, \gamma(1) = \theta'$, such that $\int \mathcal{L}(\gamma(t))dt$ is minimized [17, 18]. A number of techniques have been proposed to find such curves $\gamma$. In this work, we use the technique proposed in [17], which parameterizes the Bezier curve with $k + 1$ bends, given by $\gamma_\phi(t) = \sum_{j=0}^{k} \binom{k}{j}(1-t)^{k-j}t^j\theta_j$ for $t \in [0, 1]$, where $\theta_0 = \theta, \theta_k = \theta'$, and $\phi = \{\theta_1, \ldots, \theta_{k-1}\}$ are trainable parameters of additional models, defining "bends" on the curve $\gamma_\phi(t)$. We use Bezier curves with three bends ($k = 2$). Given the curve $\gamma_\phi(t)$, we define the mode connectivity of the models $\theta, \theta'$ to be

$$\text{mc}(\theta, \theta') = \frac{1}{2}(\mathcal{L}(\theta) + \mathcal{L}(\theta')) - \mathcal{L}(\gamma_\phi(t^*)), \tag{4}$$

where $t^*$ maximizes the deviation $t \mapsto |\frac{1}{2}(\mathcal{L}(\theta) + \mathcal{L}(\theta')) - \mathcal{L}(\gamma_\phi(t))|$. There are three possibilities for mode connectivity. If $\text{mc}(\theta, \theta') < 0$, then $\frac{1}{2}(\mathcal{L}(\theta) + \mathcal{L}(\theta')) < \mathcal{L}(\gamma_\phi(t^*))$, which means there is a "barrier" of high loss between $\theta, \theta'$; in this case, we will say that the loss landscape is poorly-connected or simply say that mode connectivity is *poor*. If $\text{mc}(\theta, \theta') > 0$, then this implies a curve of low loss connecting $\theta, \theta'$, but it also implies that the training failed to locate a reasonable optimum, i.e., $\mathcal{L}(\theta)$ and $\mathcal{L}(\theta')$ are large. If $\text{mc}(\theta, \theta') \approx 0$, then we will say that the loss landscape is well-connected or simply say that the mode connectivity is *good*. Note that for all the experiments except neural machine translation, we use the training error (0-1 loss) when computing mode connectivity, so that mode connectivity is always normalized to the range of $[-100, 100]$. We provide additional details

---

[2] For this work, we focus on the similarity of representations at the output layer, i.e., after the softmax is applied, although the CKA similarity can be used to compare the representations at any layer.

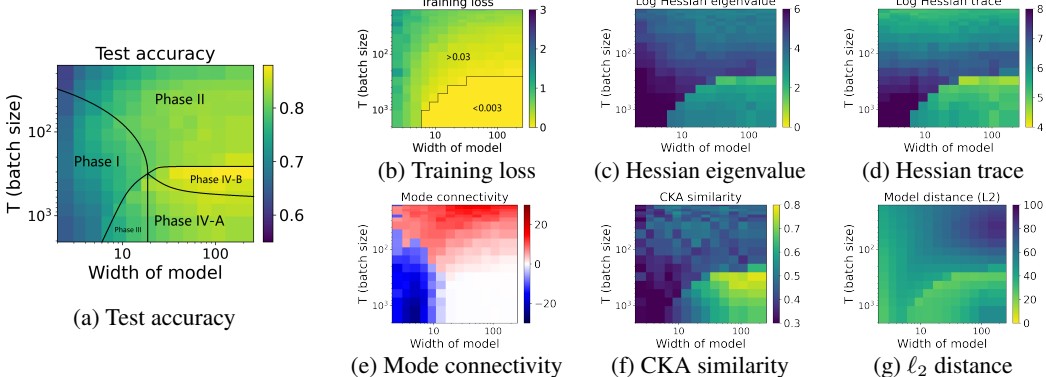

(a) Test accuracy  (b) Training loss  (c) Hessian eigenvalue  (d) Hessian trace

(e) Mode connectivity  (f) CKA similarity  (g) $\ell_2$ distance

Figure 2: **(Standard setting).** Partitioning the 2D load-like—temperature-like diagram into different phases of learning, using batch size as the temperature and varying model width to change load. Models are trained with ResNet18 on CIFAR-10. All plots are on the same set of axes. We note that batch size is inverse temperature, and thus it has smaller values at the top of the y-axis and larger values at the bottom.

on this procedure, as well as an ablation study on different mode connectivity hyperparameters, in Appendix A.4.2 of the full paper [25].

**Hessian.** The Hessian at a given point $\theta_0$ in parameter space is represented by the matrix $\nabla_\theta^2 \mathcal{L}(\theta_0)$. To summarize the Hessian in a single scalar value, we report the dominant eigenvalue $\lambda_{\max}(\nabla_\theta^2 \mathcal{L}(\theta_0))$ and/or the trace $\mathrm{tr}(\nabla_\theta^2 \mathcal{L}(\theta_0))$, calculated using the `PyHessian` software [11].

$\ell_2$ **distance.** We will also occasionally report the $\ell_2$ distance between two parameter configurations $\|\theta - \theta'\|_2$ as a measure of similarity between models, although we typically find that the CKA similarity is a more informative measure.

## 3  Empirical results on taxonomizing local versus global structure

In this section, we present our main empirical results. Among other things, our results will highlight the presence of globally nice, globally well-connected/poorly-connected, and locally flat/sharp loss landscapes, and the phase transitions which separate them. In addition to test accuracy, results on six other metrics are presented, including training loss, leading Hessian eigenvalue, trace of Hessian, CKA similarity, mode connectivity, and $\ell_2$ distance measured between model weights. For each metric, the results are presented in a 2D diagram, in which the horizontal dimension is the load (with increasing load to the right), and the vertical dimension is the temperature (with increasing temperature to the top).

We will illustrate our main results in a simple setting, and then consider several variants of this setting to illustrate how these results do or do not change when various parameters and design decisions are modified. To start, we will consider ResNets [28] trained on CIFAR-10 [29] as the standard setting to demonstrate different loss landscapes. We will scale the network width to change the size of the network. For ResNet18 which contains four major blocks with channel width $\{k, 2k, 4k, 8k\}$, we select different values of $k$ to obtain ResNet models with different widths. In the standard setting, batch size, learning rate, and weight decay are kept constant throughout training to study interactions between temperature-like parameters, load-like parameters, and the loss landscape. Below, we will apply learning rate decay and consider other variations of this standard setting, in separate experiments. More details on the experimental setup can be found in Appendix B of [25].

### 3.1  Types of loss landscapes and phase transitions

In this subsection, we discuss our standard setting, in which we vary model width as the load-like parameter and batch size as the temperature-like parameter. A summary of the results is displayed in Figure 2. Each pixel represents a specific training configuration tuple (width, batch size), averaged over five independent runs. Observe that there are two phase transitions (identified by different metrics) that separate each plot into four primary regions (corresponding to those shown in Figure 1).

- **Hessian distinguishes locally sharp versus locally flat loss landscapes.** The first phase transition is displayed in Figure 2c and 2d, separating Phase I/II from Phase III/IV. A larger Hessian eigenvalue or Hessian trace (darker color) represents a sharper local loss landscape [5, 11]. In Figure 2b, we find this transition coincides with a significant decrease in the training loss. Indeed, the training loss experiences a more than tenfold reduction when transitioning from the upper side to the lower side on the right of the figure. Comparing Figures 2a and 2c-2d, categorizing loss landscapes based solely on the Hessian (or, from other results, other local flatness metrics) is insufficient to predict test accuracy, e.g., the test accuracy in Phase III is lower than Phase IV-A but the Hessian eigenvalues are almost the same.
- **Mode connectivity distinguishes globally well-connected versus globally poorly-connected loss landscapes.** The second phase transition is shown in Figure 2e. The white region represents near-zero mode connectivity which, according to our definition, implies a flat curve in the loss landscape between freshly-trained weights; the blue region represents negative mode connectivity which implies a high barrier between weights; and the red region represents positive mode connectivity which implies a low-loss curve between weights, although the weights are not trained to a reasonable optimum. The loss along individual mode connectivity curves can be found in Appendix A.5 of the full version [25]. In contrast to training loss, test accuracy only appears to show significant improvements after this transition. In particular, for well-connected loss landscapes, one can improve the test accuracy with suitable choice of temperature. This phase transition forms a curve separating Phase I from II, and separates Phase III from IV.

Based on the two transitions, we now classify the loss landscapes into the following phases.

- **Phase I: Globally poorly-connected and locally sharp**: Training loss is high; Hessian eigenvalue and trace are large; and mode connectivity is poor.
- **Phase II: Globally well-connected and locally sharp**: Training loss is high; Hessian eigenvalue and trace are large; and mode connectivity is poor because the trained weights fail to locate a reasonable minimum.
- **Phase III: Globally poorly-connected and locally flat**: Training loss is small; Hessian eigenvalue and trace are small; yet mode connectivity still remains poor.
- **Phase IV: Globally well-connected and locally flat**: Training loss is small; Hessian eigenvalue and trace are small; and mode connectivity is good (near-zero).

We remark that in Figure 2 (and subsequent figures below) the load-like and temperature-like parameters are on the X and Y axes, respectively, and we have, to the extent possible, kept other control parameters (in particular, those which are also load-like and temperature-like) fixed, so as to isolate the effect of load-like and temperature-like behavior on trained models. One might wonder (or even criticize our experimental setup, if one were not to realize that we are trying to isolate the effects of load-like and temperature-like parameters) what would be the effect of varying learning rate (which is another temperature-like parameter) during the training process. Thus, we include the setting with decaying learning rate during training in Section 3.2.

Here are two additional observations we can make from Figure 1.

- **CKA further distinguishes two subcategories in Phase IV.** From Figure 2f, CKA can be used to further divide Phase IV into Phase IV-A and Phase IV-B, with the latter exhibiting larger CKA similarity. We remark that the transition from Phase IV-A to Phase IV-B is more like a smooth crossover than a sharp transition. Thus, we name both of them Phase IV.
- **Simple $\ell_2$ distance is not enough.** A challenge in measuring similarity between models is that the same model can be realized using different weights [8]. To reconcile this effect, the distance between two models is commonly defined in terms of their predictions instead of weights. Indeed, the representation-based CKA similarity is seen to be preferable to the weights-based $\ell_2$ distance. For example, from Figure 2g, the $\ell_2$ distance provides some limited information, but it is not as informative as CKA similarity.

Based on these results, we assert the following central claim of this work: **optimal test accuracy is obtained when the loss landscape is globally nice and the trained model converges to a locally flat region; and we can diagnose these different phases in the load-like–temperature-like phase diagram with Hessian, mode connectivity, and CKA metrics.** Importantly, both similarity and connectivity metrics are required for a globally nice loss landscape. Phase IV-B is precisely the region with *globally nice* landscapes, exhibiting the highest test accuracies.

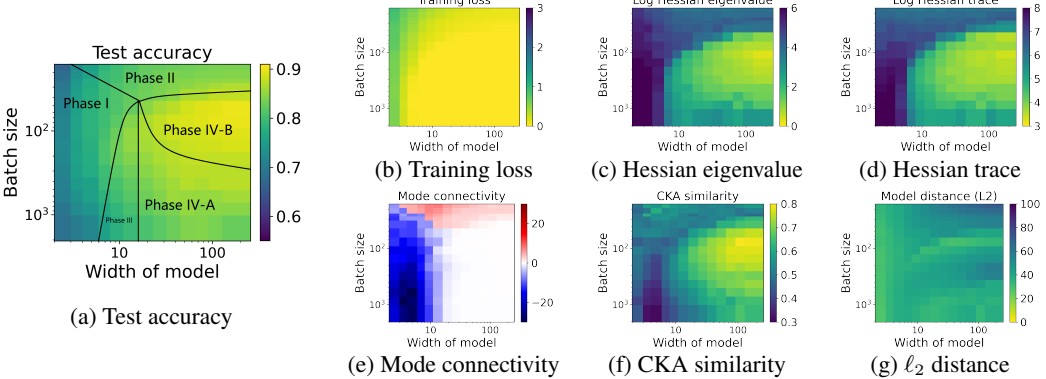

Figure 3: **(Learning rate decay).** Partitioning the 2D load-like—temperature-like diagram into different phases of learning, varying batch size to change temperature and varying model width to change load. Learning rate decay is applied during training. Models are trained with ResNet18 on CIFAR-10. All plots are on the same set of axes.

## 3.2 Corroborating results

In this subsection, we consider initial corroborating results, modifying the setup of Section 3.1 to train with learning rate decay, or to data with exogenously-introduced noisy labels, etc. Still more results can be found in Section 3.3 and in the Appendix of the full paper [25].

**Training with learning rate decay.** Next, we consider a similar experimental setup and the same phase diagram, except with the same learning rate decay schedule applied in the middle of training rather than with a fixed learning rate throughout. We still vary batch size to change temperature. The results are presented in Figure 3. Comparing Figure 3 with Figure 2, we see that the four phases are still present, and the test accuracy is maximized when the loss landscape is globally nice and locally flat. Therefore, our central claim is unaffected by the learning rate decay schedule. In Figures 3c and 3d, smaller temperatures (or larger batch size) in Phase IV-A appear to increase the size of the Hessian. This is a well-known issue with large-batch training [3]. Finally, note that the optimal test accuracy achieved improves in the presence of learning rate decay.

**Training to noisy labels and double descent.** Next, we consider a similar experimental setup and the same phase diagram, except that we randomize 10% of the training labels (similar to [30]). The results are presented in Figure 4. Comparing with Figure 2, we see that our main conclusion still holds, i.e., the loss landscape which is both globally nice and locally flat achieves the best test accuracy, shown in Phase IV-B. However, an additional observation can be made: if we compare Figure 4a with Figure 2a, a "dark band" arises between different learning phases. In particular, from Figure 4a, we see that the test accuracy exhibits both width-wise and temperature-wise double descent [21–24, 30], for certain parameter choices. In particular, the shape of the dark band matches that of the transitions shown in Figure 4c and 4d.

**Double descent and phases of learning.** The significance of this "dark band" is the following. A central prediction when viewing different phases of optimization landscapes from a statistical mechanics perspective [12, 13] is that there should be "bad fluctuations" between qualitatively different phases of learning (e.g., see the transition that separates Phase I and II from Phase III and IV in Figure 4a). The connection between phases and fluctuations in the popular double descent [21, 22] was made precise in analyzable settings [23, 24]. Here, we complement [23, 24] by exhibiting the same type of transitions empirically between different phases in our taxonomy, and demonstrating that empirical double descent is a consequence of qualitatively different phases of learning.

**Training to zero loss.** Next, we use Figure 4 to discuss whether to train to (approximately) zero loss, which is popular in recent work. From Figure 4b, we observe that Phase III and Phase IV achieve almost exactly zero loss, while Phase I and Phase II do not. Once again, the loss experiences a more than tenfold decay when transitioning from Phase I/II to Phase III/IV. However, if we restrict to globally poorly-connected regions and we restrict to a particular width value, i.e., selecting one column slice in the diagram that cuts through Phase III, such as the red block shown in Figure 4a, we see that the best test accuracy is obtained in Phase I/II, instead of Phase III. Note that Phase I/II not only does not achieve zero loss, but it also has locally sharp minima (observed from Figure 4c and

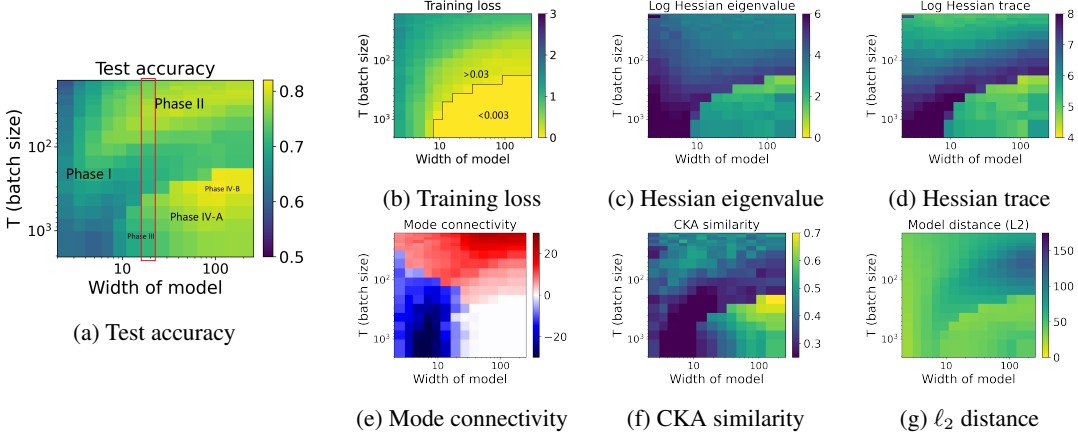

Figure 4: **(Training to noisy labels and double descent).** Partitioning the 2D load-like—temperature-like diagram into different phases of learning, using batch size as the temperature and varying model width to change load. 10% of labels are randomized, and double descent is observed between different phases. For an arbitrary column slice that cuts through Phase III (e.g., the red block), optimal accuracy is achieved in Phase I/II with locally sharp minima. Models are trained with ResNet18 on CIFAR-10. All plots are on the same set of axes.

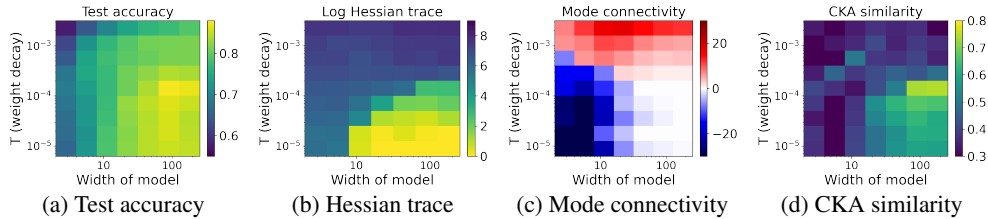

Figure 5: **(Weight decay as temperature).** Partitioning the 2D load-like—temperature-like diagram into different phases of learning, using weight decay as the temperature and varying model width to change load. Models are trained with ResNet18 on CIFAR-10. All plots are on the same set of axes.

4d). This means that **for globally poorly-connected loss landscapes, it is possible that converging to a locally flat region achieves lower accuracy than a locally sharp region.** More interestingly, this locally sharp region does not even converge to close-to-zero training loss. Thus, one will wrongly predict that Phase III outperforms Phase I/II if one only looks at local sharpness.

### 3.3 Ablation study

**Different temperature parameters.** First, we study weight decay as an alternative temperature parameter, in addition to batch size. We change the temperature parameter from batch size used in Figure 2 to weight decay, and we report the results in Figure 5. The results shown in Figure 5 are similar to those seen in Figure 2. One observation is that, once again, the best test accuracy is obtained when the loss landscape is both globally nice and locally flat. Another observation with Figure 5b is that when training a wide model with small weight decay (which is shown on the bottom of the figure), the Hessian trace becomes extremely small. This matches observations in [14] that decreasing weight decay reduces the size of the Hessian. Since weight decay is known to improve generalization, this also demonstrates that local metrics alone are insufficient to predict test performance.

**Different amount of training data.** Next, we vary the amount of training data (as another way of changing load) and see how that affects our results. We vary the number of training samples in CIFAR-10 by a factor of ten. Results are shown in Figure 6. Again, the optimal test accuracy is achieved when the Hessian eigenvalue and trace are small, mode connectivity is near-zero, and CKA similarity is large. Perhaps unsurprisingly, better test accuracy is achieved with more data. Here, CKA provides useful complementary information to the Hessian and mode connectivity for explaining the utility of larger data. The Hessian alone cannot predict the correct trend, as it *increases*

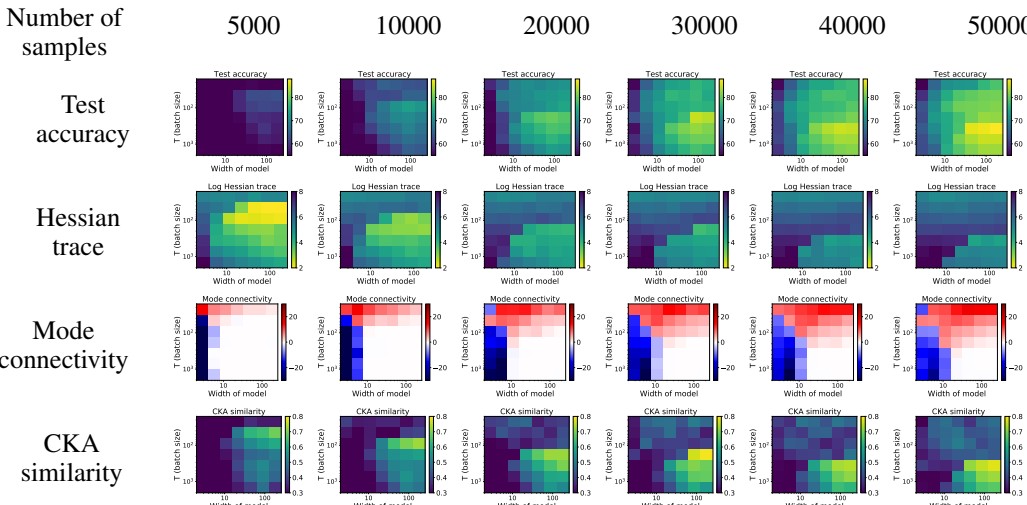

Figure 6: **(Varying amount of training data).** Partitioning the 2D load-like—temperature-like diagram into different phases of learning, using batch size as the temperature and varying model width to change load. We vary quantities of training data from CIFAR-10 in different columns. All plots are on the same set of axes.

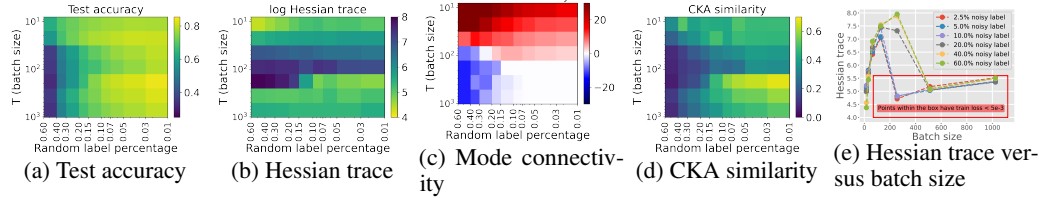

(a) Test accuracy    (b) Hessian trace    (c) Mode connectivity    (d) CKA similarity    (e) Hessian trace versus batch size

Figure 7: **(Proportion of randomized labels as load).** Partitioning the 2D load-like—temperature-like diagram into different phases of learning, using batch size as the temperature and varying proportion of randomized training labels to change load. Models are trained with ResNet18 on CIFAR-10. All plots are on the same set of axes. (e) shows that the Hessian trace changes slowly with the proportion of noisy labels when training loss is small.

in magnitude with data. Mode connectivity alone cannot predict the correct trend either, becoming increasingly poor with larger data (see the shrinking white region). Indeed, it appears that larger models are required to keep the loss landscape well-connected with increasing data. In contrast, CKA precisely captures the relationship of increasing test accuracy with additional data.

These observations also imply that the utilities of extra data and larger models are different: larger models can increase connectivity in the loss landscape (e.g., Figure 2e); while increasing data boosts signal in the landscape, enabling trained models to become more similar to each other. Clearly, researchers have been increasing both the size of data and the size of models in recent years; our methodology suggests obvious directions for doing this in more principled ways.

**Different quality of data by changing the amount of noisy labels.** Next, we vary the proportion of randomized labels to simulate the change in the *quality* of data, as another way to change load. To generate randomized labels, a percentage of the training data is randomly selected and altered to an incorrect target class. Results are shown in Figure 7. Once again, local information alone fails to measure the quality of training data. We can see that training with a large amount of noise does not significantly affect the Hessian — see Figure 7b. In particular, as the temperature decreases (down to the bottom of Figure 7b), the Hessian becomes smaller, independent of the quantity of noisy labels. This is especially evident in Figure 7e, where we plot Hessian trace against batch size. However, looking instead at mode connectivity in Figure 7c and CKA in Figure 7d, one can easily deduce that training with more noisy labels leads to more poorly-connected loss landscapes.

**Different datasets, architectures, load/temperature parameters, and training schemes.** We have performed a wide range of other experiments, only a subset of which we report here. These additional experiments can be found in the Appendix of the full paper [25]. In Appendix D, we cover additional

datasets, including SVHN, CIFAR-100, and IWSLT 2016 German to English (De-En) (a machine translation dataset), as well as additional NN architectures, including VGG11 and Transformers. While there are many subtleties in such a detailed analysis (several of which point to future research directions), all experiments support our main conclusions. Here, we briefly summarize these results.

In Appendix D.3, we study an analogous plot to Figure 4, training with 10% noisy labels but replacing the temperature-like parameter from batch size to learning rate. Again, we observe the double descent phenomenon. Using this experiment, we infer that the decision to train to zero loss (traditionally a rule-of-thumb in computer vision tasks, although note that recent work has highlighted how the difference between exactly zero versus approximately zero can matter [31]) should depend on the global connectivity of the loss landscape. Indeed, for small models with poor connectivity, we find that training to zero loss can harm test accuracy. This suggests that the common wisdom to fit training data to zero loss is derived from experiments involving relatively high-quality data and models, and is not a principle of learning more generally.

In Appendix D.4, we show that in the setting of machine translation, the loss landscape remains poorly-connected (i.e., the mode connectivity remains negative) even for a reasonably large embedding dimension up to 512 (see Figure 19). In this case, generalization can be quite poor when training to zero loss. This conclusion matches (with hindsight) the observations in practice, e.g., dropout and early stopping can improve test loss [32, 33]. It also suggests that an embedding size of dimension 512 (for six-layer Transformers with eight attention heads used in our experiments) is still not large enough for baseline machine translation, and that certain (different) training schemes should be designed to improve the optimization on these loss landscapes.

In Appendix D.5, we study learning rate as an alternative temperature parameter, which produces analogous results to Figure 2. In Appendix D.7.1, we study large-batch training and show that it increases local sharpness. Note that for most experiments, we intentionally keep a constant learning rate when varying the batch size to study the change in the landscape with a changing temperature; thus, in Appendix D.7.2, we provide additional results on tuning learning rate with changing batch size, including the commonly used "linear scaling rule" [34].

## 4  Conclusions

Motivated by recent work in the statistical mechanics of learning, we have performed a detailed empirical analysis of the loss landscape of realistic models, with particular attention to how properties vary as load-like and temperature-like control parameters are varied. In particular, local properties (such as those based on Hessian) are relatively easy to measure; and while more global properties of a loss landscape are more challenging to measure, we have found success with a combination of similarity metrics and connectivity metrics. This complements recent work that uses tools from statistical mechanics and heavy-tailed random matrix theory, as we can perform large-scale empirical evaluations using metrics (CKA, mode connectivity, Hessian eigenvalues, etc.) that are more familiar to the ML community. We interpreted these metrics in terms of connectivity and similarity, and we used them to obtain insight into the local versus global properties of NN loss landscapes.

Here, we summarize a few observations from our connectivity and similarity plots (that we expect will be increasingly relevant as larger data sets and models are considered). i) A larger width improves mode connectivity; ii) more data improves similarity; iii) better data quality improves mode connectivity; iv) a larger width and a higher temperature in Phase IV improves similarity. These observations can be restated in a way to guide the operations in training:

- A negative mode connectivity suggests that the data quality is low or the model size is small.
- A large positive mode connectivity or a large Hessian leading eigenvalue/trace indicates that the training fails to converge to the bottom of a local minimum.
- A small CKA similarity suggests that generalization is not good, which can be caused by various factors. However, if, in addition, the mode connectivity is close to zero, and the Hessian is small, a large CKA similarity indicates the "lack of signal" from the data. In other words, one should get more high-quality data for training.

In future work, we aim to provide a more detailed study on using the metrics for improved training, and we will look at phase diagrams outside of the load/temperature form, especially in the low-connectivity regime, which is most challenging according to our taxonomy.

**Acknowledgements.** We want to thank Charles Martin, Rajiv Khanna, Zhewei Yao, and Amir Gholami for helpful discussions. Michael W. Mahoney would like to acknowledge the UC Berkeley CLTC, ARO, IARPA (contract W911NF20C0035), NSF, and ONR for providing partial support of this work. Kannan Ramchandran would like to acknowledge support from NSF CIF-2007669, CIF-1703678, and CIF-2002821. Joseph E. Gonzalez would like to acknowledge supports from NSF CISE Expeditions Award CCF-1730628 and gifts from Alibaba Group, Amazon Web Services, Ant Group, CapitalOne, Ericsson, Facebook, Futurewei, Google, Intel, Microsoft, Nvidia, Scotiabank, Splunk and VMware. Our conclusions do not necessarily reflect the position or the policy of our sponsors, and no official endorsement should be inferred.

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
