# OpenReview forum: "Taxonomizing local versus global structure in neural network loss landscapes"
_NeurIPS.cc/2021/Conference — NeurIPS 2021 Poster_

### Official Review · Reviewer_guMx · 2021-07-15

**Rating:** 8
**Confidence:** 4

**Summary:**

The author(s) perform an empirical study of which (and how) properties of the landscape affect the test accuracy, obtaining results that are consistent over a large number of deep neural network models over several credible datasets. The study puts into practice this idea that the generalization is not uniquely determined by local properties of the landscape, but rather also by global properties. They argue that different kinds of acting on the system and on the dynamics can be grouped into two control parameters that they call 'load' and 'temperature', and show that according to the position in the (load, temperature) plane one has different phases. They observe that each phase seems related to a different test accuracy, and realize that to obtain the best test accuracy, in addition to being locally flat, the landscape should also be globally "nice". By "nice", they mean that the landscape is well-connected (no large barriers separating regions) and that different trained models have a large CKA similarity.


**Limitations And Societal Impact:**

One limitation that I see is that the temperature and load axes are (perhaps deliberately) defined in a fuzzy manner. They do this to emphasize that different drivers have qualitatively similar effects. I am not sure that all the drivers that are collected under the same umbrella actually represent the same kind of behavior (e.g. weight decay as temperature). I don't see this as a problem, since these issues fall out of the scope of the paper and can be addressed and understood in future work.




**Main Review:**

The paper addresses a long-due question, trying to explain that local flatness cannot be used as a sole indicator of a good minimization. They address this question numerically, through large-scale experiments over a wide range of parameters. The results seem quite robust across architectures and datasets. The main message of the paper is simple and compelling, and will likely have a good resonance throughout the community.

I do have a series of concerns and comments, however, that I would like the authors to address.

I will mostly refer to Fig.2, but many of the comments apply to several figures.

Despite being conceptually very simple, the results along the temperature axis are extremely confusing. This is  due to the way labels, axes and captions are presented. The y axis of Fig.2 is labeled "temperature (batch size)". However, the batch size is the inverse temperature. So, is the quantity on the y axis the batch size or its inverse? Further, the direction of the axis is reversed (bigger values are below), which makes everything even more confusing. Finally, it took me a long time before I realized that the axis was inverted, because the fonts are so small that they can bearly be read with a 200% zoom (and impossible to read when printed).
So what is the quantity on the y axis? Is zero temperature at the top or bottom (an intuitive figure would put it at the bottom)? I think that in the y axis temperature goes upwards and batch size goes downwards. This makes sense but it took me quite a while to understand the figure, and the reader should be able to grasp it immediately.
The captions and labels should clearly explain what is plotted, and help the reader in the interpretation.
What do the subplot titles in Fig2 mean? I read batch size, and I assume that it is because the temperature parameter here is the batch size. If so, this is redundant (thus confusing), since it is already written on the y axis. However, the same "batch size" labels appear in figure 3, where the caption says that this figure is "using learning rate as temperature". Yet further confusion is caused by the main text, which instead states that a learning rate decay was used. In this case, the authors rightly state that the temperature is not well-defined. Besides the contradiction with the figure caption, I assume then that what the authors are stating is that the y axis stands for the initial temperature value. This independence from the learning rate is interesting, though I would be more conservative, because in many complex systems the rate of annealing (how fast the learning rate decreases) can be crucial to this effect [].


I do not understand why Fig.2e reveals a transition on a vertical line. By looking at the figure I see a vertical line in the bottom part of the plot, but this line then curves and possibly even splits. How are the authors identifying this phase transition?

The authors talk of phases, but the word phase has a specific meaning related to a non-analyticity in a (free) energy functional. For sharpness and connectivity a phase transition seems possible (actually, it was shown in https://arxiv.org/abs/1809.09349 that the one along the load axis is actually a phase transition), but the one from phase IV-A to phase IV-B seems more a smooth crossover, since I see no sharp change in the CKA similarity nor in any other observable. Am I right?

It seems to me that the CKA similarity alone is enough to explain the test accuracy (the two quantities correlate extremely well), without the need of the other metrics. Can the authors elaborate on this?

In all shown examples, the interpolation threshold (where the training loss becomes zero) seems to be at a higher load than the MC connectivity. From Ref https://arxiv.org/abs/1809.09349, which identifies a jamming transition at the interpolation threshold, I would have expected the two to coincide. Can the authors say anything on this? Can it be due to the way the MC was computed (both because of the method and the k=2 choice)?

In the supplemental material the authors state that most experiments were run until convergence, but they stopped at 150 epochs when the convergence criterion was not met. I think that this can influence the results, so I would least specify in the main text which runs did not reach convergence. Does this apply to the runs (and only those) with a negative mode connectivity?

The authors talk about "poor" mode connectivity, but this term is not defined. Is a poor MC positive? Is it non-zero (so also negative)?

Line 206. The sentence is not clear, also due to the "poor" term.

The results are shown for VGG and Resnets. I wonder whether they also hold for simpler models such as multi-layer perceptrons and simply convolutional networks, since these models are more tractable analytically. Moreover, can the authors say anything about linear models?

Would the authors have found equivalent results if they studied the mode connectivity through the Nudged Elastic Band method, as done in http://proceedings.mlr.press/v80/draxler18a/draxler18a.pdf?

Fruitful examples of the study of loss landscapes can be found in https://www.annualreviews.org/doi/pdf/10.1146/annurev-conmatphys-031119-050745, from random landscapes, to jamming, to spin glasses. Ref http://proceedings.mlr.press/v80/draxler18a/draxler18a.pdf (cited elsewhere in the text) is a great example.
When mentioning sharp vs flat minima, I would have expected a citation to https://arxiv.org/abs/1609.04836.
The concept of temperature is not new as well as the interpretation in terms of a Gibbs distribution. See e.g. https://arxiv.org/pdf/1711.04623.pdf and references therein.
The concept of load is not new either (except for the name).

On line 295 they can cite e.g.
- Q. Li, C. Tai, and E. Weinan, Stochastic modified equations and adaptive stochastic gradient algorithms,inInternational Conference on Machine Learning(PMLR, 2017) pp. 2101–2110.
- W. Hu, C. J. Li, L. Li, and J.-G. Liu, On the diffusion approximation of nonconvex stochastic gradientdescent, Annals of Mathematical Sciences and Applications4(2019), arXiv:1705.07562.
- Y. Zhang, A. M. Saxe, M. S. Advani, and A. A. Lee, Energy–entropy competition and the effectivenessof stochastic gradient descent in machine learning, Molecular Physics116, 3214 (2018)

Anyhow, I am not sure that section 4 adds any knowledge at all. The authors have a picture in their mind, and they draw it schematically, in one way out of infinitely many that they could have chosen. It might even be wrong, since it assumes the existence of only two relevant scales of ordering (local and global), while they could be infinite (e.g. with spin glasses, jamming, ...). The concept of effective landscape already exists, and it is called Free Energy. This quantity comprehends the competition between energy (loss) and thermal agitation, it can in principle be obtained by legendre-transforming the energy (= the loss), and was studied for decades (one reference is the just given Zhang et al paper, but we can even resort to papers from the 80s from Gardner and coworkers).

There is something awkward with the batch indices at line 116. The concept is simple and clear, but the authors introduce some names of indices which it is not clear to what they correspond unless one already knows what the authors mean.

Line 162: which typically occurs if training failed to locate a reasonable optimum. This statement is true for a strict inequality, but the authors are referring it to mc<=0.

Line 338: "we provide theoretical explanations". I disagree. The paper is purely empirical and this is fine.

Line 344: the statement seems naif.


**Time Spent Reviewing:**

9

---

> ### Author Response · Authors · 2021-08-10
> **Response to R4 (guMx)**
>
> We sincerely thank the reviewer for such insightful and detailed comments. Here are our responses.
>
> ### Making figures. Typo in the caption of Figure 3.
>
> We have already revised our paper based on your comments. See below.
> - We now point out (explicitly, in the main text and the figure caption) that all the y-axes have a large temperature on top and a small temperature at the bottom. Thus, batch size, which is inverse temperature, has large values at the bottom and small values at the top.
> - We have removed the "batch size" (and other temperature parameters) from the title of each subplot to avoid confusing redundant information.
> - We have now increased the text size in the figure to the largest possible without affecting the typesetting.
> - We apologize for the typo in the caption of Figure 3. In this figure, learning rate decay is applied during training. However, batch size is still the temperature parameter that we change. Thus, the caption should be "varying batch size to change temperature" instead of "using learning rate as temperature". Note that only difference between Figure 3 and Figure 2 is that the experiments in Figure 3 apply learning rate decay, while those in Figure 2 do not. Therefore, in Figure 3, it is hard to define temperature since the learning rate decays during training. However, all the training processes in Figure 3 use the same learning rate schedule, and we can still use batch size to measure the (relative) temperature.
>
> Here are the new captions respectively for Figure 2 and Figure 3.
>
> - Figure 2: (Standard setting). Partitioning the 2D load-like—temperature-like diagram into different phases of learning, using batch size as the temperature and varying model width to change load. Models are trained with ResNet18 on CIFAR-10. All plots are on the same set of axes. We note that batch size is inverse temperature, and thus it has smaller values at the top of the y-axis and larger values at the bottom.
> - Figure 3: (Learning rate decay). Partitioning the 2D load-like—temperature-like diagram into different phases of learning, varying batch size to change temperature and varying model width to change load. Learning rate decay is applied during training. Models are trained with ResNet18 on CIFAR-10. All plots are on the same set of axes. We note that batch size is inverse temperature, and thus it has smaller values at the top of the y-axis and larger values at the bottom.
>
> ### Poor mode connectivity. Line 206.
>
> First, we would like to clarify that there is some inconsistency in our definition in the main paper and the definition in the appendix. The definition in Eq.(4) should be the following:
> $\textsf{mc}(\theta,\theta') = \frac{1}{2}(\mathcal{L}(\theta)+\mathcal{L}(\theta')) - \mathcal{L}(\gamma_\phi(t^\ast))$.
>
> Thus, there are two cases when MC is poor. The first is when MC has a large positive value. This happens when the trained models $\theta$ and $\theta'$ fail to locate at a reasonable local minimum. The second is when MC has a large negative value. This happens due to the big barrier. MC is good only when it is close to 0. In Figure 2e, the only part with good MC is the white region. The red and the blue regions are both poor but for different reasons. The MC has a large positive value for the red region, and it has a large negative value for the blue region.
>
> ### Phase transition with incomplete vertical line.
>
> While making Figure 2e, our choice in drawing the line between Phases I and II was largely informed by the (more fine-grained) plot shown in Figure 11 that exhibits the individual MC curves. We note by MC curve, we mean the curve of training error on the path found using the technique in [15]. See Figure 11 for details. The verticle line in Figure 2a highlights the fact that the "barrier" in the middle of a MC curve is not affected significantly by the change of temperature. This is true because, for example, all the curves on the fourth column in Figure 11 show a barrier, while the barriers on the fifth column become invisible. The vertical line that we drew in Figure 2a was intended to show this phenomenon. We do acknowledge that there is some difficulty in determining exactly the boundary between Phase I and II using Figure 2e alone; this issue may be an inevitable shortcoming of summarizing a MC curve with a single number. We will include a remark to clarify this.
>
> ### Transition from Phase IV-A to phase IV-B is smooth.
>
> We agree. This is also why we do not use Phase IV and Phase V but use Phase IV-A and IV-B. We want to use two "phases" to highlight that even for a well-connected global loss landscape, subtlety can arise due to the shape of the loss curve. We will include a remark to discuss this.
>
> ### CKA alone can explain the test accuracy.
>
> CKA does provide the best correlation with test accuracy among the metrics that we study. Nonetheless, our intention is not only to give a single-number generalization metric. The collection of global and local metrics tell much more. Please refer to the answer to R3 (4CJS) on "Empirical guidance from the phase diagram". The main takeaway there is that different metrics can be used to suggest different operations for training and data processing.
>
> ### The jamming and the MC transition do not coincide.
>
> First, we can confirm that the mismatch between the MC and the jamming transition is not due to hyperparameters. See Figure 10 in the appendix.
>
> Second, based on results, we believe that the interpolation point does not necessarily coincide with the MC transition: two interpolating models need not be well-connected. Indeed, connectivity consistently requires higher load (in particular width/overparameterization) than does interpolation.
>
> ### Stopping at 150 epochs can influence the results that do not converge.
>
> For training processes that do not meet this criterion, we have increased our training epoch to 150 to ensure that they enter a "steady" stage. The only cases where the training loss still slowly decreases and have not converged are those shown in Figure 17, on which we have explicitly marked the ones that do not converge. We do not see a clear connection between the hardness to converge (e.g., the region in Fig 2b that is not bright yellow) and the barrier on the mode connectivity curve (e.g., the blue region in Fig 2e).
>
> ### Simpler networks.
>
> We have indeed conducted similar experiments on smaller models, such as two-layer convolutional networks. The major issue is that these nets are usually hard to train to close-to-zero training loss. Thus, it is hard to show a complete picture as Figure 2a.
>
> ### Nudged Elastic Band method.
>
> We will mark that down as another method to try for finding low-loss curves. We expect the results to be similar to what we have.
>
> ### Section 4 adds no knowledge. Suggested papers.
>
> We appreciate the additional references provided by the reviewer, and we will be happy to include a more thorough comparison to these works in an updated draft of the paper.
>
> As the reviewer points out, the concepts of temperature (and its relation to the Gibbs distribution) and load are not new, and indeed much of the reason for conducting the present study is motivated by previous work in the statistical physics literature. The purpose of this work is to study how the properties of actual neural network landscapes vary as we change temperature-like and load-like parameters, and we believe our empirical results demonstrate that this perspective can bear useful insights.
>
> The particular form of the toy model used is indeed _designed_ to match the schematic presented in Figure 1. The purpose of defining the simple toy model and the effective landscape is primarily to help develop intuition for how the behavior illustrated in Figure 1 can arise, in particular as the temperature parameter varies. Our definition of the effective landscape does capture much of the same information as the free energy, though, from our perspective, in a more visually appealing way. Nonetheless, the reviewer's point is well-received, and we will consider carefully how to better present the material in this section in an updated draft of the paper.
>
>
>
> ### Other comments.
>
> We will address accordingly.

---

> > ### Comment · Reviewer_guMx · 2021-08-31
> > **Satisfied with comments**
> >
> > Thanks for the response. I think that after addressing the stated points the manuscript will be highly improved.
> >
> > I would still remove section 4, in favor of more thorough explanations in the rest of the sections, such as explaining the choice of the verticality of the MC line, which in my view still seems somewhat arbitrary.
> >
> > Everything else seems fine.

---

### Official Review · Reviewer_4CJS · 2021-07-16

**Rating:** 6
**Confidence:** 3

**Summary:**

This paper explores factors that influence the loss landscape. In particular, it categorizes the loss landscape in different phases, each corresponding to different status of key factors such as Hessian spectrum and mode connectivity. It also tries to establish an "effective loss landscape" model that depends on key parameters.

**Limitations And Societal Impact:**

 The authors addressed the limitations and potential negative societal impact of their work.

**Main Review:**

Exploring key factors that affect the global landscape and training has been a hot research area. Many work has focused on the influence of one particular factor (such as Hessian, connectivity, etc). The advantage of this work is that it combines several factors together and looks at the comprehensive influence. I believe a single factor is not enough to explain all about landscape and training, and a phase diagram is a good trial along multi-factor direction. Another interesting point is the category of load and temperature-like parameters. It seems that different load/temperature-like parameters have similar effect on the landscape. This may help unify the influence of different hyper-parameters.

The following are my questions and concerns about this paper.

1. This work shows the phase transition of some key parameters. What is the cause of the transition, and is it possible that these parameters are still intermediate parameters that depend on some other key factors?

2. Hessian spectrum and CKA similarity perform similarly on most results but different in a few. What is the essential difference between these two factors, and what standard should we choose if we have to choose one?

3. Is there any empirical guidance this phase diagram can provide? The empirical results show the landscape and key parameters at the end of the training. Is it possible that we acknowledge the current phase during early iterations?

4. I'm a bit confused about the effective loss landscape. I'd like to see more about why defining effective loss landscape is reasonable (and necessary) for observing the influence of temperature and load.

5. The explanation of (4) when discussing mode connectivity is contradictory in the main paper and the appendix. In the main paper, it claims that there is a barrier if $mc>0$, while in the appendix both the description and the figures show that there is a barrier when $mc<0$.

**Time Spent Reviewing:**

6 hours

---

> ### Author Response · Authors · 2021-08-10
> **Response to R3 (4CJS)**
>
> We appreciate the reviewer for giving us positive feedback and offering us the chance to clarify some important aspects, especially for those on training and empirical guidance. Please find our response below.
>
> ### 1. Key factors that cause the phase transition. Key parameters versus intermeduate parameters.
>
> First of all, we want to clarify that Hessian, mode connectivity and CKA are tools to measure the phases of learning. The design of the tools themselves is not the core point of this paper. Second, the temperature-like and load-like parameters are important factors that can change the loss landscape. However, these are not the key parameters mentioned by the reviewer. In fact, we select these temperature-like and load-like parameters to study something more fundamental, which are changes in the global and local structure of the loss landscapes. Thus, we view the global and local structure as the underlying factors that lead to phase transitions. We consider the temperature-like and load-like parameters as intermediate parameters that can help us conduct systematic studies.
>
> ### 2. Hessian and CKA perform similarly. What standard should we use to choose between the two?
>
> The major difference between Hessian and CKA is in Phase III and IV-A. Phase III is the one with locally flat and globally poorly-connected loss landscapes. Please note that Phase III in Figure 2 is the one with the smallest font at the bottom of the figure. In this phase, using Hessian alone predicts good generalization, but CKA predicts lower generalization performance. In Phase IV-A, we observe a globally well-connected but less convex-like loss landscape compared to Phase IV-B. In this phase, Hessian alone predicts good generalization, but CKA predicts that generalization performance is not good enough. See Figure 2c and 2f for the difference.
>
> As for choosing between the two, our tentative suggestion is to always use CKA when the computation power allows. As we have argued in the Introduction, global metrics are harder to measure than local metrics. Still, they provide more information than local ones and are less likely to provide confounding information.
>
> ### 3. Empirical guidance from the phase diagram
>
> We have given empirical guidance in our conclusions. In the following, we would like to restate them in a way that can better guide the operations in training.
> - A negative mode connectivity suggests that the data quality is low or the model size is too small.
> - A large positive mode connectivity or a large Hessian leading eigenvalue/trace suggests that the training fails to converge to the bottom of a local minimum.
> - A small CKA similarity suggests that generalization is not good, which can be caused by various factors. However, if in addition, the mode connectivity is close to zero, and the Hessian is small, a large CKA similarity indicates the "lack of signal" from the data, i.e., one should buy more data for training.
>
> ### 3. (Continued from the question above.) Acknowledging phases early in training.
>
> One can indeed measure mode connectivity (MC) during training. However, in this case, one should look at the whole MC curve instead of the single-value metric shown in the phase plots. If the global loss landscape is poorly connected, the MC curve will show a large barrier in the middle. At the same time, the two endpoints will also be high because the training has not converged. The resulting plot will look like the top-left corner of Figure 11 in the appendix, which shows the individual MC curves trained on a poorly connected loss landscape with a large temperature. On the other hand, CKA and Hessian can provide more reasonable results at the end of training than during training because, during training, Hessian and CKA are likely to be high. We note that Hessian is a relatively well-studied topic compared to global landscape metrics. For example, see [4] for one example of using Hessian to tune hyperparameters during training.
>
> ### 4. Defining effective loss landscape for observing the influence of temperature and load.
>
> The purpose of defining the effective landscape is to primarily demonstrate the effect of varying temperature on the effective landscape explored during optimization. Since the actual loss function being optimized does not change as parameters of the optimization algorithm (e.g. batch size, learning rate) vary, the concept of the effective landscape is important for understanding the taxonomy presented in Figure 1. In particular, our definition of the effective landscape captures the fact that, depending on the temperature used during optimization, the optimizer will concentrate more or less heavily on different regions of the loss surface. The definition is particularly inuitive in one dimension, and can be easily visualized, hence the focus on that setting.
>
> ### 5. Definition of poor mode connectivity.
>
> We thank the reviewer for the careful comment. Indeed, we have used large negative values to denote poor mode connectivity with barriers, i.e., MC<0 when there is a large barrier. In other words, the definition in the appendix is the correct one. We have revised our paper to remove the conflicts in definition.

---

### Official Review · Reviewer_EVPJ · 2021-07-17

**Rating:** 6
**Confidence:** 4

**Summary:**

The paper proposed an experiment framework for evaluating loss landscape of neural networks during training. The experimental setup systematically vary two types of hyperparameters: 1) temperature type, where at high temperatures, larger learning rate or smaller batch size will induce more noise to training, and 2) load type, including data quantity/quality and model width. Neural networks are trained with different hyperparameter combinations according to the load-temperature pairs. Then 3 metrics including mode connectivity, Hessian, and output CKA similarity evaluated on perturbed training data. Jointly consider the 3 metrics (local and global) evaluated on all loss landscapes, the authors classify the loss landscapes into different phases. One of the phases (IV with high connectivity, low Hessian and high CKA) corresponds to best test accuracy.


**Limitations And Societal Impact:**

Yes

**Main Review:**

The paper jointly considers three types of metrics that were considered in literature to evaluate both global and local characteristics of loss landscapes. It explains some of the counter-intuitive phenomena observed such as why in some low hessian loss landscape, the model does not generalize well. Experiments were run with 3 types models (ResNet, VGG and Transformer) on different data sets. However comparing the results figures, ResNet vs Transformer, it is not clear if the defined phases are universally useful. On the other hand, the effect of optimizer was not discussed.
Overall, the paper is well written.

Minor comments:
In Figure 4, when evaluating ‘global nice’, global connectivity seems the only thing that matter, as highest accuracy is at 0 mode connectivity and low (but not lowest) Hessian.
Figure 3 should the y axis label be learning rate instead of  batch size, since learning rate is used as temperature? Otherwise, what do the ‘batch size’ in both subtitle and axis labels mean? The figure captions need to be more clear.

The authors clarified how to interpret the results and made corrections to figures in their responses. I am more convinced.


**Time Spent Reviewing:**

3 hours

---

> ### Author Response · Authors · 2021-08-10
> **Response to R2 (EVPJ)**
>
> Thank you for the feedback. We believe your comments are important to help us clarify some crucial points that we have not included in the main paper. Please find below our response.
>
> ### Universality of the defined phases. ResNet vs Transformer. The effect of optimizer.
>
> Thanks for the comment. First of all, we would like to bring to the attention of the reviewer that our paper does cover numerous factors that affect the neural network loss landscape. The factors that we have studied include data quantity (which we vary by 10x), data quality (which we vary by changing the amount of randomized labels from 0 to 60%), different datasets (including Cifar10, Cifar100, SVHN, and IWSLT16), different architectures (including ResNets, VGGs, and Transformers), different temperature-like parameters (including learning rate, batch size, and weight decay), and different training procedures (including training with a constant learning rate shown in Figure 2, training with learning rate decay shown in Figure 3, large-batch training shown in Figure 19, and training with the "linear-scaling rule" shown in Figure 20). The selected factors are intended to be extensive, yet it is possible that we still lack some important factors. However, we believe we've been able to cover factors that are core to our study.
>
> Next, we would like to clarify the comment about the universality of our phase diagram. The primary purpose of the phase diagram is to systematically study the difference and connections between local and global loss landscape properties. Through measuring a few quantities, including mode connectivity, CKA and Hessian, we identify the different types of loss landscapes. Thus, the main contributions are to taxonomize these different cases and point out the importance of looking at both global and local structures. We DO NOT intend to claim that the particular shape of the phase diagram remains unchanged in all different machine learning problems.
>
> Furthermore, as we have discussed in detail in Appendix E.3 and E.4, the shape of the phase diagram can be affected by multiple subtle issues. For example, given the size of the Transformer models that we can physically train in the scale of hundreds, we cannot observe Phase IV of the globally well-connected and locally flat loss landscape. We hypothesize that this is because the Transformers are not large enough. Although this issue makes the shape of the diagram in Figure 17 visually different from that observed in the computer vision results, the observation itself is interesting. It shows that machine translation, in general, is "harder" than image classification (e.g., for ResNets trained on Cifar10) because the loss landscape is not well-connected even for Transformers of medium size. Other factors, such as the double-descent phenomenon observed in Figure 16 and 17, can also significantly affect the shape of the diagram. Thus, we believe the more important thing is the taxonomy of different types of loss landscapes instead of the particular shape of the diagram.
>
> We hope the above discussion helps explain the consistency between results on ResNets and Transformers. For more details, please refer to discussions in Section E.3 and E.4.
>
> As for the effect of the optimizer, as we have already mentioned, we aim to cover different factors that help resolve the confounding factor on global versus local loss landscapes. We hope the reviewer can recognize our honest effort to present the results in a comprehensive way without withholding subtleties in the detailed analysis.
>
> ### Global connectivity seems to be the only one that matters in Figure 4.
>
> First, we agree with the reviewer that local information measured by Hessian does tend to give misleading information in Figure 4, as the Hessian becomes extremely small when measured on models trained with extremely small weight decay, despite the fact that extremely small weight decay can reduce the test accuracy. In fact, studying the confounding factors that lead to the correlation between local metrics and test accuracy is the core of our paper.
>
> Furthermore, we want to point out that CKA provides additional information to mode connectivity in finding the optimal test accuracy. In general, we believe this comment by the reviewer supports our main claim, which is that confounding factors can lead to the false belief on one-fits-all generalization metrics. Figure 4 provides an example of how local metrics alone can lead to a biased prediction of test accuracy.
>
> ### Temperature label in Figure 3. Figure caption.
>
> We apologize for this typo. The caption of Figure 3 should be "varying batch size to change temperature", instead of "using learning rate as temperature". The only difference between Figure 3 and Figure 2 is that learning rate decay is applied during the training processes shown in Figure 3, while learning rate is kept constant (for each training process) in Figure 2. In both Figures, the varied temperature parameter is batch size. Now, we have changed the caption of Figure 3 to the following.
>
> - Figure 3: (Learning rate decay). Partitioning the 2D load-like—temperature-like diagram into different phases of learning, varying batch size to change temperature and varying model width to change load. Learning rate decay is applied during training. Models are trained with ResNet18 on CIFAR-10. All plots are on the same set of axes. We note that batch size is inverse temperature, and thus it has smaller values at the top of the y-axis and larger values at the bottom.

---

> > ### Comment · Reviewer_EVPJ · 2021-08-16
> > **Thanks for the response**
> >
> > Thank you for the detailed response. I suggest the authors move some of the discussions from E.3 and E.4 to the main text, to help readers interpret the results.
> > I have increased the score accordingly.

---

> > > ### Author Response · Authors · 2021-08-16
> > > **Thanks for your quick response**
> > >
> > > We sincerely appreciate your quick feedback on our rebuttal. We have moved some of the discussions from Appendix E.3 and E.4 to the main paper.

---

### Official Review · Reviewer_qCke · 2021-07-18

**Rating:** 7
**Confidence:** 4

**Summary:**

This paper takes a large number of neural networks that span range of different parameters (e.g. width), train them in a large number of ways (e.g. dataset, batch size), and studies several metrics that characterize the local (e.g. Hessian spectral norm) and global (weight vector pair-wise connectivity on potential non-linear paths). By doing a large scale study of this kind, the authors are able to identify several distinct phases of loss landscapes, dividing them phenomenologically into (globally {well,poorly}-connected) x (locally {flat, sharp}). The authors also develop a simple mathematical toy model that can model this taxonomy.

**Limitations And Societal Impact:**

Yes

**Main Review:**

Overall, I really like this paper. I am very happy that the a large scale study of this kind has been done and that the different local and global aspects of loss landscapes of DNNs have been investigated in this way. I enjoyed reading the paper, the message is clear, and so are the experiments. I have a number of points of confusion and clarifications but overall I'm a fan! Even if the paper potentially doesn't get accepted I'd strongly suggest the authors put it on arXiv so that the community can benefit from their results.

STRONG POINTS:
1) a clear theoretical backing of the questions
2) a well written paper
3) looking at both local and global aspects of the loss landscape (the global aspects are often understudied)
4) a nice mathematical toy model allowing us to conceptualize the taxonomy the authors present

WEAKER POINTS AND QUESTIONS:

1) In your experiments you vary a lot of parameters such as the network width, batch size, etc. Once you change a hyperparameter, the optimal value of the other hyperparamters also changes. This is certainly the case when you're spanning orders of magnitude as you do. In your results and plots, is each datapoint the optimal set of hyperparameters for that specific choice of the controlled hyperparameters, or are all of them kept constant. For example, let's say you have batch size, learning rate, width, and L2 regularization, and let's say you're looking at mode connectivity vs the learning rate. Are you, for a specific choice of learning, choosing a new, optimal (through a grid search let's say) values of (batch size and L2 regularization). If not, this could be a serious confounder for your experimental conclusions. I think this point as the key question I have that needs to be addressed.

2) In the introduction, you mention that the best test accuracy is achieved when the ensembles of trained models are more similar to each other. This seems to go both against 1) the received wisdom that ensembles work well because of function diversity, 2) many theoretical results in this direction, and 3) empirical work on this (for example see https://arxiv.org/abs/1912.02757 which might be a relevant reference for you as well, since they also deal with mode connectivity a lot). How do you reconcile your conclusion with the previous results?

3) Your toy model is one dimensional? Some papers argue that it's the high dimensional nature of the loss landscape that plays a key role in forming many of its properties. For example https://arxiv.org/abs/1906.04724 (NeurIPS'19) looks at a similar question to you but with a distinctly high-D lens. Does the predictive power of your toy model depend on its dimension, or are you just trying to capture the relevant phenomenology with 1-D?

4) Bigger and better models? The results you show are on the weaker side of CIFAR-10 test accuracies. In Figure 2 it seems that the color scale doesn't even go to 90% test acc. It might be better to add stronger, better optimizer models, and at least CIFAR-100 as well to make the case for your taxonomy stronger.

5) I am a bit worried that the large "phase changes" you see in e.g. Figure 2 Panel e) are caused by the really bad losses you have there. You use cross entropy loss on 10 classes, so the default value should be ln(10) ~ 2.3. But your barriers are often of the order ~10. How is that possible? And would the transitions look as sharp if you didn't have these really high losses to compare to?

RELEVANT PAPERS
I also thought of a few more relevant papers.
- On large scale structure of loss landscapes: https://arxiv.org/abs/1906.04724
- On ensembles from the loss perspective: https://arxiv.org/abs/1912.02757
- On linear mode connectivity and even init - optimum connectivity from an unrelated init: http://proceedings.mlr.press/v139/lucas21a.html
- On the large scale structure of the Tr(H): https://arxiv.org/abs/1807.02581
Feel free to add them if you see that they are relevant, but no pressure for sure.

Overall I really like this paper! It could be made better by potentially addressing the suggestions I made above, but as it is it's certainly valuable to the community and should get published one way or another (and arXived for sure as soon as the authors see fit).

UPDATE AFTER REBUTTAL:
I am sorry for being slightly late into the reply game. I am very happy with the answers the authors provided and would like to keep my score of 7 = accept. I think this paper is a valuable contribution to the subfield. Good job!

**Time Spent Reviewing:**

4

---

> ### Author Response · Authors · 2021-08-10
> **Response to R1 (qCke)**
>
> We sincerely thank the reviewer for the insightful and constructive feedback. We've addressed all of your comments below.
>
> ### 1. When studying the effect of one hyperparameter, do you tune other hyperparameters?
> In all except one of our results, we kept other hyperparameters constant when changing one. We understand the importance of hyperparameter tuning, and we would like to restate why we intentionally kept other hyperparameters constant. We will also discuss the one exception where we do tune the other hyperparameters.
> - Tuning more than one hyperparameter confounds load and/or temperature. A primary purpose of this paper is to show how load and temperature-like parameters affect the loss landscape by putting them in a 2D grid. By keeping other hyperparameters constant, moving along one of the axes correlates directly to a change in load or temperature. However, if we tune, for example, learning rate and batch size altogether, we will not know how the load or temperature changes.
> - Please note that we have considered the issue of hyperparameter tuning prior to our submission. In Appendix E.6.2, we have an example of tuning learning rate and batch size together using the "linear scaling rule". As we have discussed on line 285-288, we designed this experiment to specifically study the practice of tuning learning rate together with batch size. However, in Figure 20 of Appendix E.6.2, we do not see any significant change along the y-axis. This is expected because tuning the learning rate and batch size together using the linear scaling rule yields similar temperature regardless of the batch size. Thus, this way of tuning two hyperparameters at the same time makes it difficult to study how temperature and/or load affect the loss landscape.
>
> ### 2. Ensembles work well because of function diversity.
>
> We want to bring to the attention of the reviewer that the setup of our work is different from Bayesian neural networks and deep ensembles. Indeed, in Bayesian neural networks, it is useful to make the trained networks diverse such that they cover different modes of the posterior. However, our setting is different in that we want the trained models to be more similar to each other as measured by the CKA similarity to reflect the fact that, in this sense, the models come from a concentrated posterior. We will cite the paper on deep ensemble in the revised paper.
>
> ### 3. One-dimensional toy model.
>
> Indeed, the purpose of the 1-D toy model is to demonstrate only qualitatively how varying temperature can alter the "effective" landscape explored during optimization, and to justify the taxonomy laid out in Figure 1. Of course, the toy model is not intended to model the complexities of high-dimensional landscapes (of which there are many), but rather to help formalize this concept.
>
> ### 4. Bigger and better models.
>
> Two ways of training can improve the test accuracy on Cifar10.
> - First, using learning rate decay during training can increase the test accuracy to 90+(See Figure 3 in the main paper which uses learning rate decay). We appologize that there is a typo in the caption of Figure 3, where "using learning rate as temperature" should actually be "varying batch size to change temperature". The setting of Figure 3 is almost exactly the same as in Figure 2 except that learning rate decay is applied.
> - Second, using data augmentation can further improve test accuracy to near SOTA. However, as we have stated in line 852-855 of the appendix (which is a section that gives the details of our training procedures), we found that data augmentation can make training with noisy labels hard to converge. Thus, we decided not to use data augmentation in any of our experiments to avoid possible confounding factors. However, we do think that different types of data augmentation procedures can provide interesting ways to adjust load (if viewed as effective amount of data) and temperature (if viewed as data-dependent regularization), which could become meaningful future work to extend the current paper.
>
> Finally, we do have an experiment on Cifar100 in Section E.1. Again, we believe that for Cifar100, adding learning rate decay can provide a similar effect as Figure 3, which studies learning rate decay, and adding data augmentations to Cifar100 can improve the test accuracy to near SOTA. We will add a remark to clarify this point.
>
> ### 5. Phase transitions are caused by bad losses.
>
> We thank the reviewers for carefully reviewing our paper. We have missed an important clarification before introducing the mode connectivity plot, which is that for almost all of the plots on mode connectivity (except the one on NLP), we used the 0-1 loss so that the mode connectivity results are normalized to a range of [-100, 100]. A significant portion of our trained models shown on Figure 2 do have close-to-zero cross-entropy training loss (e.g., see the light yellow area in Figure 2b). In particular, the entire Phase III has close-to-zero cross-entropy training loss, which is to the left of the transition between globally poorly-connected and well-connected loss landscapes. Thus, the phase transition is not caused by poor training. We want to thank the reviewer again for helping us clarify this point. We have now included a note on 0-1 loss in the revised paper.
>
> ### 6. Suggested papers.
>
> Thanks for suggesting these references. We will make sure to include them in the revised version.

---

> > ### Comment · Reviewer_qCke · 2021-08-31
> > **A respose to a response**
> >
> > I am sorry for being slightly late into the reply game. I am very happy with the answers the authors provided and would like to keep my score of 7 = accept. I think this paper is a valuable contribution to the subfield. Good job!

---

### Author Response · Authors · 2021-08-10
**Response to all the reviewers**

We want to thank all the reviewers for constructive feedback, which helps us improve our paper. We want to sincerely thank the first (qCke) and the fourth (guMx) reviewers for providing us with detailed comments. We also want to thank the second (EVPJ) and third (4CJS) reviewers for offering us the chance to clarify several important aspects of the paper. Please find below the response to each comment.

---

### Decision · Program_Chairs · 2021-09-27

**Decision:**

Accept (Poster)

**Comment:**

This paper investigates how the structure of the loss landscape of neural networks affects their generalization performance.  The authors categorize loss landscapes into four different phases, globally {well,poorly}-connected) x (locally {flat, sharp}), where global connectivity is measured by mode connectivity and local flatness is measured using Hessian. Through a large-scale empirical investigation of neural network loss landscapes, the authors present an analysis of how these categories affect generalization.

Overall, I think this is a solid paper. The reviewers provided a lot of useful detailed feedback and the authors also did a great job of addressing the reviewer concerns. During the discussion phase, the consensus decision leaned towards acceptance.

I recommend acceptance and encourage the authors to address the reviewer comments in the final revision.